# Necrotic Bone Fluid Suppresses Energy Metabolism of Porcine PBMC-Derived Macrophages In Vitro

**DOI:** 10.3390/cells14161258

**Published:** 2025-08-14

**Authors:** Zhuo Deng, Chau P. Nguyen, Yan Liu, Jaehyup Kim, Thomas P. Mathews, Chi Ma, Yinshi Ren, Chao Xing, Harry K. W. Kim

**Affiliations:** 1Center of Excellence in Hip, Scottish Rite for Children, Dallas, TX 75219, USA; judy.deng@tsrh.org (Z.D.); chau.nguyen@tsrh.org (C.P.N.); chi.ma@tsrh.org (C.M.);; 2Eugene McDermott Center for Human Growth and Development, University of Texas Southwestern Medical Center, Dallas, TX 75390, USA; 3Department of Pathology, University of Texas Southwestern Medical Center, Dallas, TX 75390, USA; 4Children’s Research Institute, University of Texas Southwestern Medical Center, Dallas, TX 75390, USA; thomas.mathews@utsouthwestern.edu; 5Department of Orthopaedic Surgery, University of Texas Southwestern Medical Center, Dallas, TX 75390, USA

**Keywords:** Legg–Calvé–Perthes disease (LCPD), osteonecrosis (ON), macrophage, necrotic bone fluid (NBF), energy metabolism

## Abstract

Legg–Calvé–Perthes disease is a juvenile ischemic osteonecrosis (ON) of the femoral head. A disruption of blood supply to the femoral head produces extensive cell death and necrotic debris. Macrophages are innate immune cells recruited to the necrotic bone to orchestrate the repair process. However, the role macrophages play in the ON repair process is still not elucidated. The purpose of this study was to determine the effect of artificial necrotic bone fluid (NBF) on porcine peripheral blood mononuclear cell (PBMC)-derived macrophages. Monocytes were positively selected by CD14 MicroBeads from pig PBMCs. After maturation, cells were treated with no stimulant (Con), LPS + IFNγ (M1), IL4 + IL13 (M2), or NBF. All culture supernatants and cells were harvested for ELISA, Western blot, FACS, RT-qPCR and bulk RNAseq. The Western blot and ELISA showed that only the M1 condition elevated the protein level of pro-inflammatory cytokines. The FACS results indicated that percentage of CD8086+ (M1 marker) cells was significantly lower in the M2 vs. other conditions, whereas the relative median fluorescence intensity of CD8086 was significantly higher in the M1 vs. other conditions. The NBF did not show any significant change compared to the Con. mRNA analysis showed significantly increased IL1β and IL8 expression in the M1 vs. Con scenario. TNFα expression was significantly decreased in the M2 vs. Con scenario. Interestingly, the NBF did not induce pro-inflammatory gene expression. For bulk RNAseq, the Gene Set Enrichment Analyses of the M1-stimulated cells revealed the enrichment of pro-inflammatory gene sets. For the M2, most of the enriched categories were related to the down-regulation of inflammation. For the NBF, the most enriched categories were related to the down-regulation of protein translation and mitochondrial metabolism. We further confirmed the suppressive effects of NBF on macrophage functions using Seahorse Cell Mito Stress Tests, ^13^C-glucose metabolic flux analysis, mitochondrial ROS detection via MitoSOX^TM^ staining, and phagocytosis assay. Taken together, these results revealed that the artificial NBF down-regulates the overall cellular activity and energy metabolism of macrophages.

## 1. Introduction

Legg–Calvé–Perthes disease (LCPD) is a juvenile ischemic osteonecrosis (ON) of the femoral head [1,2]. Chronic inflammation is a consistent pathological feature of LCPD patients, which impairs bone healing and regeneration [3,4]. A disruption of blood supply to the femoral head severely restricts the supply of oxygen and nutrients, which produces extensive cell death and the abundance of necrotic cell debris in the bone marrow space and the trabecular bone. The dying cells and necrotic debris leave damage-associated molecular patterns (DAMPs) in the necrotic bone microenvironment and impair bone healing [5,6]. We observed that DAMPs in the necrotic bone fluid (NBF) significantly inhibited osteoblast differentiation and stimulated the fibroblast differentiation of mesenchymal stem cells (MSCs) [7]. 

Following ischemic ON, innate immune cells are the primary modulators of early inflammatory response [8]. However, how innate immune cells respond to DAMPs and regulate inflammation in the necrotic bone environment has not been fully elucidated. Previous studies on DAMP-induced inflammation have been limited to non-bone tissues. These studies show that DAMPs trigger inflammatory response from various innate immune cells by activating multiple signaling pathways through the pattern recognition receptors [9,10,11]. Macrophages are cells that respond early to tissue damage and play an essential role in the innate immune response to bone injury and healing [4,12]. Thus, our study sought to elucidate the effects of NBF on macrophages, which are central innate immune cells recruited to the necrotic bone. 

Macrophages are known to have wide phenotypic plasticity, ranging from a pro-inflammatory phenotype (M1, classically activated macrophages) to an anti-inflammatory phenotype (M2, alternatively activated macrophages), depending on the local microenvironment [13,14]. In vitro, like human and mouse cells, the porcine macrophage pro-inflammatory phenotype can be induced by LPS and IFNγ, while anti-inflammatory phenotype can be polarized by IL4 and IL13 [15,16]. The pro-inflammatory macrophages secrete massive pro-inflammatory cytokines and lead to the amplification of inflammatory responses; anti-inflammatory macrophages release anti-inflammatory cytokines and perform efferocytosis to promote tissue healing and repair [4,17,18]. 

The coordinated activity of M1 and M2 macrophages is crucial for proper bone healing following injury [19]. M1 macrophages dominate the early phase by initiating inflammation and recruiting additional immune cells and MSCs through pro-inflammatory cytokines such as TNF-α and IL-1β [19,20]. This pro-inflammatory phase is followed by a phenotypic shift toward M2 macrophages, which resolve inflammation, secrete tissue-repair cytokines like IL-10 and TGF-β, and stimulate osteogenesis and angiogenesis [20,21]. M2 macrophages actively promote bone repair through phagocytosis of apoptotic and necrotic cells, a process essential for resolving inflammation and enabling regenerative signaling in bone marrow environments [8,22,23]. Impaired M2 phagocytic activity within necrotic bone marrow, as seen in osteonecrosis, may thus delay healing and promote chronic inflammation [18]. A balanced M1/M2 transition is therefore essential for effective resolution of inflammation and bone regeneration. However, disruption of this balance—such as sustained M1 activation or impaired M2 polarization—can contribute to pathological outcomes. In ischemic osteonecrosis, prolonged M1-like responses and insufficient M2-mediated repair mechanisms may perpetuate chronic inflammation, delay healing, and exacerbate bone degeneration [18]. 

Macrophage function is closely linked to their energy metabolism, which plays a critical role in determining their phenotype and activity. Pro-inflammatory M1 macrophages rely mainly on glycolysis, a rapid but inefficient energy pathway that supports cytokine production and reactive oxygen species (ROS) generation. In contrast, anti-inflammatory M2 macrophages depend on mitochondrial oxidative phosphorylation (OXPHOS) and fatty acid oxidation, which support tissue repair and resolution of inflammation [24,25]. These metabolic programs not only reflect the activation state of macrophages but also regulate their behavior, making immunometabolism a key control point in inflammation and healing [26,27].

Previous immunohistochemical studies of necrotic femoral heads from both human and pig demonstrate macrophage infiltration in the necrotic bone marrow and surrounding synovium [6,28,29,30]. However, more in-depth analysis of the effects of necrotic bone microenvironment on macrophage phenotype and response have not been performed. Since the porcine model of ischemic ON of the femoral head is a well-established model of LCPD and shows similar histologic and radiographic features to the human disease [31], we isolated and cultured porcine peripheral blood mononuclear cells (PBMCs) to investigate the effects of NBF on macrophage phenotype. The purpose of this study was to obtain transcriptomic characterization of porcine PBMC-derived macrophages cultured with NBF. We used artificially created NBF, as described previously [7], and M1- and M2-stimulated macrophages served as pro- and anti-inflammatory references. We hypothesized that NBF will produce pro-inflammatory response from porcine macrophages. 

## 2. Materials and Methods

### 2.1. Experimental Design

The animal protocol was approved by the Institutional Animal Care and Use Committee (IACUC) at University of Texas Southwestern Medical Center (UTSW). A total of 8 Yorkshire male pigs were used. The whole blood from 5 pigs were used for ELISA, Fluorescence-activated cell sorting (FACS), RNA analyses, and metabolic flux analyses and from 3 pigs for the Seahorse Cell Mito Stress Test to evaluate macrophage metabolic activity and cell fluorescence staining. All blood was collected in blood collection bags with heparin as anti-coagulant. Artificial NBF was produced from normal pig distal femoral epiphyseal bone, as pre-viously described [7], and stored in −80 °C until use. 

### 2.2. PBMC Isolation and Culture

Ficoll-Paque PLUS density gradient media (Cytiva, Marlborough, MA, USA) was used to isolate PBMCs from the whole blood. The “buffy layer” was collected after the gradient centrifugation [32]. After washed three times by phosphate-buffered saline (PBS) (Thermo Fisher Scientific, Waltham, MA, USA), monocytes were positively selected by human CD14 MicroBeads (Miltenyi Biotec, Bergisch Gladbach, Germany) using magnetic activated cell sorting (MACS) method. More than 96% of CD14+ cells were confirmed by FACS analysis (CD14 antibody: BioRad, Hercules, CA, USA) after the MACS selection (Appendix A). CD14+ monocytes were seeded into 6-well Petri dishes as 5 × 10^6^ cell/well, and cultured in RPMI 1640 medium (Thermo Fisher Scientific) containing 1% porcine serum (PS) (Thermo Fisher Scientific) and 1% Penicillin/Streptomycin (Thermo Fisher Scientific). The plates were pre-coated with PS in 37 °C incubator for at least one hour. After 4 days of culture in 37 °C incubator for macrophage differentiation and maturation, cells were differentiated from monocyte to macrophages as most cells were firmly attached to the cell culture plate. Cells were then treated with no stimulant (Control, Con), M1 stimulants (LPS 100 ng/mL+ IFNγ 20 ng/mL), M2 stimulants (IL4 20 ng/mL + IL13 20 ng/mL), or artificial NBF (100 µg/mL). The artificial NBF was produced following the previous protocol [7]. Briefly, distal femurs were collected from piglets immediately after euthanasia under sterile conditions. After removing the articular cartilage and growth plate, the bony epiphyses were subjected to five freeze–thaw cycles (liquid nitrogen for 3 min, then 37 °C water bath for 30 min). The bones were cut into ~5 mm^3^ pieces and washed with saline (2:1, saline:bone, *v*/*w*). After 5 min incubation and 1 min vertexing, the mixture was filtered through a 40 μm strainer (Corning, Corning, NY, USA), centrifuged at 1500 rpm for 10 min, and the supernatant was sterilized using a 0.22 μm Millex filter (Millipore, Burlington, MA, USA). The final NBF was stored at −80 °C until use. All cells were harvested 24 h after the stimulation for flow cytometry (FACS), and RNA isolation. The cell culture medium supernatants were also collected, aliquoted, and froze in −80 °C for enzyme-linked immunosorbent assays (ELISA).

### 2.3. Western Blot

Cells were lysed using RIPA buffer (Thermo Fisher Scientific) supplemented with Halt^TM^ Protease Inhibitor Cocktail (100×) (Thermo Fisher Scientific) to extract total protein. The protein concentrations were determined using Pierce^TM^ BCA Protein Assay Kit (Thermo Fisher Scientific). The same amount of protein (40 µg) of each condition were separated by SDS-PAGE and transferred onto PVDF membranes (Millipore). Membranes were detected by primary antibodies β-actin (Cell Signaling Technology, Danvers, MA, USA), IL-1β (R&D Systems, Minneapolis, MN, USA), and TNFα (Abcam, Cambridge, UK). Odyssey Imaging System (LI-COR Biosciences, Lincoln, NE, USA) was used to visualize the signal produced by IRDye secondary antibodies against the corresponding primary antibodies. 

### 2.4. ELISA

Porcine IL-1 beta/IL-1F2 Quantikine ELISA Kit (R&D Systems) and Porcine TNF-alpha Quantikine ELISA Kit (R&D Systems) were used to measure the expression level of IL1β and TNFα in the cell culture supernatants. The procedure followed the manufacturer’s instructions. Briefly, the secreted proteins were detected by conjugated IL1β and TNFα antibodies, respectively. The substrates recognized the conjugated antibody and released optical signals. The signals were read at 450 nm by POLARstar Omega Plate Reader (BMG Labtech, Ortenberg, Germany).

### 2.5. FACS Sample Preparation

For FACS analysis, cells were harvested using Macrophage Detachment Solution (PromoCell, Heidelberg, Germany). Macrophages were stained with anti-CD152 antibody (also known as CD8086, M1 marker) conjugated with PE (Ancell, Richmond, Australia) and anti-CD203a antibody (also known as SWC9) conjugated with Alexa Fluor^®^ 647 (BioRad). All stained samples and non-stained control (NS) were run in LSR II FACS machine (BD Biosciences, Franklin Lakes, NJ, USA) in Flow Cytometry Core in UTSW. The FACS data were analyzed in FlowJoTM v10 software (BD Biosciences). The gating strategy was shown in Appendix A. 

### 2.6. RNA Isolation and RT-qPCR

Total RNA of all samples were extracted from the 6-well plates using PureLinkTM RNA Mini Kit (Thermo Fisher Scientific). The amount of 500 ng of total RNA of each sample was employed to synthesize cDNA using iScriptTM Reverse Transcription Supermix (BioRad) for RT-qPCR. The mRNA expression of the interested genes were measured using iTaq Universal SYBR Green Supermix (BioRad) in a QuantStudio 6 Flex qPCR machine (Thermo Fisher Scientific). The primers used in this study were shown in Table 1 [33,34,35].

### 2.7. Bulk RNA Sequencing and Analysis

The amount of 3 µg of total RNA of each sample was also submitted to the Next Generation Sequencing Core at UTSW for bulk RNA sequencing (RNAseq). RNA integrity and quality was measured by Agilent BioAnalyzer 2100 (Agilent Technologies, Santa Clara, CA, USA), the RNA integrity number (RIN) of all samples were close and above 8.0 (Appendix A). A total of 12 libraries (4 groups, *n* = 3) were prepared and sequenced in paired-end mode (150 PE) with the depth of 35–45 M reads per sample (Illumina, San Diego, CA, USA). The differentially expressed gene (DEG) was performed using edgeR [36], and identified between the M1 vs. Con, M2 vs. Con, and NBF vs. Con. Gene Set Enrichment Analyses (GSEA) were performed using the WebGestalt website [37,38]. The DEGs and their rank scores (ranked based on the sign of fold change and the number of *p*-values) were input into the WebGestalt. Both human and porcine gene ontology (GO) databases were applied to annotate the DEGs. We further analyzed the top 500 DEGs (based on *p*-value and all *p* < 0.05) of each pair using Metascape website [39], which integrated multiple databases to annotate genes and perform pathway enrichment and network analyses.

### 2.8. Seahorse Mito Stress Test

To measure the mitochondrial respiration and glycolysis levels, Seahorse Cell Mito Stress Test was applied. All supplies for the test came together as the Seahorse XF Cell Mito Stress Test Kit (Agilent Technologies), and all procedures followed the manufacturer’s instructions. Briefly, CD14+ monocytes were seeded into the Seahorse XF96 Tissue Culture Microplates as 4 × 10^5^ cell/well. Cells were cultured and treated as the same conditions described above. Cells were ready for the test 24 h after the stimulation, and washed three times using Seahorse medium (DMEM (Sigma D5030, Thermo Fisher Scientific) supplemented with 1% Pen/Strep, 2 mM glutamine, 1 mM pyruvate, and 10 mM glucose; adjusted to 7.4 pH). The cells were then incubated in the Seahorse medium for 1 h at 37 °C in a non-CO_2_ chamber. Three drugs were loaded in the pre-hydrated cartridge accordingly to modulate the mitochondrial respiration by interfering the complexes of the Electron Transport Chain (ETC) during the assay. Oligomycin is an inhibitor of ATP synthase complex V, which inhibits mitochondrial respiration but leads to maximal glycolysis; carbonylcyanide-3-chlorophenylhydrazone (CCCP) is an uncoupling agent to depolarize mitochondrial membrane potential and leads to maximum mitochondrial respiration; Antimycin-A is an inhibitor of complex III. Oxygen consumption rate (OCR), and extracellular acidification rate (ECAR) were measured using Seahorse XFe96 Analyzer (Agilent Technologies). After the test, the OCR and ECAR data were normalized to the relative viable cell number, which was measured using CellTiter 96^®^ AQueous One Solution Cell Proliferation Assay (Promega, Madison, WI, USA).

### 2.9. Metabolic Flux Anlysis

To assess metabolic flux through central carbon metabolism, macrophages were cultured in glucose-free RPMI medium (Gibco, Waltham, MA, USA) supplemented with 5% dialyzed fetal bovine serum (FBS) and 2 g/L [U-^13^C_6_] glucose (Cambridge Isotope Laboratories, Tewksbury, MA, USA) for isotope tracing. Cells were incubated under their respective conditions (Con, M1, M2, or NBF) for 2 h before metabolite extraction. For intracellular metabolite collection, cells were washed twice with cold saline and quenched with ice cold 80% methanol. Cells were then scraped, transferred to microcentrifuge tubes, and frozen and thawed three cycles between liquid nitrogen and 37 °C water bath. Lastly, the tubes were centrifuged at 14,000× *g* for 10 min at 4 °C, and the supernatants were collected and kept in −80 °C. The samples were then delivered to UTSW Metabolomics Facility at Children’s Medical Center Research Institute for GC-MS service [40,41]. The incorporation of ^13^C into metabolites such as pyruvate, citrate, and others was determined by monitoring isotopologue distributions (e.g., pyruvate m+3, citrate m+2) using appropriate mass windows. Metabolite enrichment was quantified and normalized to glucose m+6 levels for relative flux comparisons among different conditions. 

### 2.10. MitoSOX^TM^ Red Live Cell Imaging and Quantification

To assess mitochondrial superoxide production, cells were stained with MitoSOX™ Red mitochondrial superoxide indicator (Thermo Fisher Scientific) according to the manufacturer’s protocol. Briefly, CD14+ macrophages were seeded into a 96-well cell culture treated plate (Corning) as 1.4 × 10^5^ cell per well. Cells were cultured and treated similarly as described above. After 24 h, the medium was replaced with 100 µL 500 nM MitoSOX^TM^ red reagent working solution. Cells were then incubated for 10 min at 37 °C in the dark. Following incubation, cells were gently washed once with Hank’s Balanced Salt Solution (HBSS) to remove excess dye. Hoechst (Thermo Fisher Scientific) was used to stain the nuclei. Live cell fluorescence images were captured immediately using Zeiss AXIO Observer fluorescence microscope (Zeiss, Oberkochen, Germany) with appropriate filters for MitoSOX™ Red (Ex/Em: 510/580 nm) and Hoechst (Ex/Em: 350/461 nm). MitoSOX^TM^ fluorescence intensities were quantified using ImageJ software 1.54 g and normalized to the number of nuclei stained with Hoechst, allowing comparison of mitochondrial ROS levels across all conditions. 

### 2.11. Phagocytosis Assay

Phagocytic activity was assessed using the Phagocytosis Assay Kit (IgG FITC) (Cayman Chemical, Ann Arbor, MI, USA), following the manufacturer’s protocol. Briefly, cells were seeded into a 96-well cell culture-treated plate and subjected to the same culture and treatment conditions as described above. At 24 h post-treatment, the medium was replaced with FITC-IgG-coated particles diluted 1:400 in fresh medium. Cells were incubated at 37 °C for 1 h in a cell culture incubator to allow the phagocytosis process. After incubation, non-internalized particles were gently removed by washing with cold PBS. To quench extracellular fluorescence, cells were treated with Trypan Blue according to the kit instructions. Nuclei were counterstained with Hoechst. Live-cell fluorescence images were captured using a fluorescence microscope equipped with appropriate filters for FITC (Ex/Em: 495/519 nm) and Hoechst (Ex/Em: 350/461 nm). Phagocytic activity was quantified as the mean FITC fluorescence intensity normalized to the number of Hoechst-stained nuclei using ImageJ software. 

### 2.12. Statistical Analysis

GraphPad Prism v10 (GraphPad Software, San Diego, CA, USA) was applied to visualize data and perform statistical analysis. One-way ANOVA with post hoc Tukey’s multiple comparison tests was used to compare 4 culture conditions. A *p*-value < 0.05 was considered statistically significant.

## 3. Results

### 3.1. Cellular Cytokine Expression and Secretion

To compare the protein levels of pro-inflammatory cytokines, we measured the expressions of IL1β and TNFα in both cell lysates and cell culture supernatants among the Con, M1, M2, and NBF conditions. We found that the expression of both IL1β and TNFα in cell lysate were higher in M1 condition compared to other conditions (Figure 1A). We did not observe a significant increase in released IL1β expression among the 4 conditions (Figure 1B). However, the secreted TNFα level in the M1 condition was significantly elevated compared to the Con (*p* < 0.0001), M2 (*p* < 0.0001), and NBF (*p* < 0.0001) conditions (Figure 1C). The results indicate that only the M1 condition elevated the expression of pro-inflammatory cytokine IL1β and TNFα by macrophages at 24 h after the stimulation, and not the M2 and NBF conditions.

### 3.2. Cell Surface Marker Expression

We compared the expressions of cell surface markers CD8086 and SWC9 among the Con, M1, M2, and NBF culture conditions, as CD8086 and SWC9 expressions are representative of pro-inflammation and macrophage maturation markers, respectively [42,43]. The percentage of CD8086+ cells in the M2 condition was significantly lower than the Con (*p* = 0.0025), M1 (*p* = 0.0003), and NBF conditions (*p* = 0.0023) (Figure 2A–E). The relative median fluorescence intensity (MFI) of CD8086 of the M1 condition was significantly higher than the Con (*p* = 0.0079), M2 (*p* < 0.0001), and NBF (*p* = 0.0019) conditions. However, the MFI of CD8086 of the M2 condition was significantly lower than the Con condition (*p* = 0.018) (Figure 2F). The results indicated that the M1 condition significantly increased the expression of pro-inflammatory marker CD8086, whereas the M2 condition significantly decreased its expression. For the macrophage maturation marker SWC9, the percentages of SWC9+ cells showed no significant difference among the 4 conditions (Figure 3A–E). However, comparison of MFI of SWC9 among the 4 conditions revealed significantly higher MFI in the M2 condition compared to the M1 condition (*p* = 0.039) (Figure 3F). Taken together, the M1 condition dramatically increased the pro-inflammatory surface marker, while the NBF condition did not increase the pro-inflammatory marker expression.

### 3.3. Pro- and Anti-Inflammatory Gene Expression

We compared the mRNA expression of multiple pro- and anti-inflammatory markers among the Con, M1, M2, and NBF conditions. The expressions of pro-inflammatory cytokines IL1β (*p* = 0.06) and IL8 (*p* = 0.0008) were significantly increased with the M1 condition compared to the Con condition (Figure 4A,B). The expressions of IL8 and TNFα were significantly higher in the M1 condition compared to the M2 (IL8: *p* = 0.0006 and TNFα: *p* = 0.0005) and NBF (IL8: *p* = 0.0017 and TNFα: *p* = 0.040) conditions (Figure 4B,C). The expression of TNFα was significantly decreased with the M2 condition compared to the Con condition (*p* = 0.0060) (Figure 4C). However, we did not observe any significant changes in anti-inflammatory gene expression (Arg1, CD163, CD206) among the 4 conditions (Figure 4D,E). These results showed that the pro-inflammatory markers were significantly induced by the M1 condition and not by the other conditions. Interestingly, NBF did not induce pro-inflammatory gene expression.

### 3.4. Transcriptomic Analysis

To comprehensively understand the gene expression changes in transcriptomic landscapes, we performed bulk RNAseq analysis and compared the 4 culture conditions. A total of 16,694 expressed gene IDs were identified after the sequencing. The Biological Coefficient Variation (BCV) plot showed the biological variability of the gene expression patterns among different samples. In the BCV plot, the samples under the same condition were separated from the samples cultured under different conditions. The 3 biological repeats of the M1 and NBF conditions were closely clustered together within the same condition, while the biological repeats of the Con and M2 conditions displayed a greater dispersion (Appendix A). The variability of the gene expression patterns within the Con and M2 conditions were greater than within the M1 and NBF conditions. Pairwise comparison between the M1 vs. Con, M2 vs. Con, and NBF vs. Con revealed specific DEGs of each comparison. The volcano plots showed the up- and down-regulated gene expression of the M1, M2, and NBF conditions compared to the Con condition. Most of the top DEGs were up-regulated with the M1 condition, while the most of the top DEGs were down-regulated with the M2 and NBF conditions (Appendix A). 

To functionally interpret the DEG results, we performed the GSEA annotated by both human and pig GO database. Comparing the results from two databases, 13,356 unique gene IDs out of all identified 16,694 expressed gene IDs were unambiguously mapped to the human database, whereas 12,844 unique gene IDs were mapped to the pig database. The GO Slim summary showed that human database annotated around 90% of unique gene IDs to the Biological Process, Cellular Component, or Molecular Function categories, while only around 30% of unique gene IDs were annotated using the pig database (Figure 5). Therefore, we applied the human GO database to perform the GO Biological Process enrichment analyses. For the M1-stimulated macrophages, most of the top enriched categories were pro-inflammatory related gene sets, such as up-regulated responses to multiple types of interferon, chemokine, molecules of bacterial origin (Figure 6A). For the M2-stimulated macrophages, most of the enriched categories were related to the down-regulation of inflammation, such as reduced responses to humoral immunity, neutrophil-mediated immunity, and granulocyte activation (Figure 6B). For the NBF condition, the top enriched categories were related to down-regulation of protein translation and mitochondrial metabolism, revealing overall down-regulation of cellular activity and energy metabolism of the cells treated with NBF (Figure 6C). In addition, we further analyzed the top 500 DEGs of each culture condition using Metascape website (Appendix A). The results validated the findings from the GSEA GO Biological Process analyses.

### 3.5. Mitochondrial Respiration and Glycolysis Profile

To verify the effects of NBF and macrophage polarization conditions on cell energy metabolism, OCR and ECAR values were measured using the Seahorse Cell Mito Stress Test. The results of the Cell Proliferation Assay did not show a significant difference in cell numbers among the 4 culture conditions (Figure 7A). The different stimulants did not affect the cell growth and viability. The normalized OCR dynamic figure illustrated that the M1 and M2 conditions increased the overall OCR compared to the control, but NBF decreased it (Figure 7B). The statistical analysis of the basal respiration levels among the 4 conditions revealed that the OCR value of the M1 condition was significantly higher than the Con (*p* < 0.0001), M2 (*p* < 0.0001), and NBF (*p* < 0.0001) conditions; the M2 condition was significantly higher than the Con (*p* < 0.0001) and NBF (*p* < 0.0001) conditions; and the NBF condition was significantly lower than the Con condition (*p* = 0.0038, Figure 7C). Similarly, for the maximal respiration level, the M1 condition was significantly higher than the Con (*p* < 0.0001), M2 (*p* = 0.0005), and NBF (*p* < 0.0001) conditions; the M2 condition was significantly higher than the NBF (*p* < 0.0001) condition; and the NBF condition was significantly lower than the Con condition (*p* = 0.002) (Figure 7D). Taken together, NBF suppressed the macrophages’ respiratory activity and capacity, which was opposite of the M1 and M2 conditions.

The comparison of basal glycolysis levels among the 4 conditions showed that ECAR value of the M1 condition was significantly higher than the Con (*p* = 0.0001), M2 (*p* = 0.0079), and NBF (*p* < 0.0001) conditions; but the NBF condition was similar to the Con condition (*p* = 0.63) (Figure 7E). However, for the maximal glycolysis level, the M1 condition was significantly higher than the Con (*p* < 0.0001), M2 (*p* = 0.0014), and NBF (*p* < 0.0001) conditions; the M2 condition was significantly higher than the NBF condition (*p* = 0.0002); and the NBF condition was significantly lower than the Con condition (*p* = 0.0004) (Figure 7F). Taken together, NBF suppressed both the mitochondrial respiration and the glycolysis capacity of the macrophages, which down-regulated the overall cellular activity and energy metabolism.

### 3.6. Metabolic Flux Profile

To further confirm the metabolic differences among the 4 conditions, we cultured macrophages in [U-^13^C_6_] glucose-containing medium and measured the incorporation of ^13^C into downstream metabolites. We analyzed the relative abundance of citrate m+2 (citrate molecules containing 2 ^13^C atoms) normalized to glucose m+6 (fully labeled glucose). Notably, M1 macrophages showed a significantly higher proportion of citrate m+2 compared to control (*p* = 0.004) and NBF conditions (*p* = 0.0001), suggesting enhanced glucose oxidation into the TCA cycle. In contrast, NBF-treated macrophages exhibited significantly lower citrate m+2 labeling compared to M2 (*p* = 0.0085), indicating suppressed mitochondrial glucose metabolism under NBF treatment (Figure 8). The result supported the notion that early M1 activation promotes active mitochondrial oxidation of glucose-derived carbon, while NBF dampens this metabolic flux. 

### 3.7. Mitochondrial Superoxide (ROS) Production

To assess mitochondrial reactive oxygen species (ROS) production under different conditions, we performed live cell imaging using MitoSOX^TM^ staining and quantified the fluorescence intensity normalized to cell number (Figure 9A,B). The results showed that M1 macrophages exhibited significantly elevated mitochondrial ROS levels compared to Con (*p* = 0.0475), indicating increased mitochondrial oxidative stress during M1 polarization. No significant differences in MitoSOX^TM^ intensity were observed in M2 or NBF-treated macrophages compared to the Con (Figure 9B).

### 3.8. Macrophage Phagocytic Activity

To assess the phagocytic activity of macrophages under different conditions, we performed a phagocytosis assay using an IgG-FITC–based detection kit. Fluorescence intensity, normalized to cell number, was quantified to compare phagocytic capacity across conditions (Figure 10). M2 macrophages exhibited significantly higher phagocytotic activity compared to M1 (*p* = 0.0287) and NBF (*p* = 0.0253) conditions. No significant difference was observed among Con, M1 and NBF conditions (Figure 10B). These results indicated that M2 polarization enhanced phagocytic activity of macrophages, whereas M1 and NBF conditions did not promote this function. 

## 4. Discussion

Understanding the impact of NBF on macrophages is crucial to elucidating the early inflammatory and immune responses within the necrotic femoral head. In this study, we discovered that artificial NBF did not polarize PBMC-derived macrophages into the pro-inflammatory phenotype as we hypothesized. Instead, NBF exerted a significant inhibitory effect on macrophage energy metabolism and cellular function. As expected, the M1 stimulants (LPS + IFNγ) acted as pro-inflammatory inducers producing significantly elevated pro-inflammatory markers at both protein and mRNA levels compared to the other conditions (Figure 1, Figure 2 and Figure 4A–C). However, the M2 stimulants (IL4 + IL13) did not trigger elevated expressions of anti-inflammatory genes and proteins, as expected, but only suppressed the expression of TNFα gene expression and pro-inflammatory surface marker CD8086 (Figure 2, Figure 3 and Figure 4C). The GESA analyses further supported these observations. For M1-stimulated macrophages, most of the top enriched categories were pro-inflammatory related gene sets, such as up-regulated responses to multiple types of interferon, chemokine, and molecules of bacterial origin (Figure 6A and Appendix A). The M1-stimulated macrophage transcriptomic profile was consistent with the transcriptomic profile of the M1-induced porcine bone marrow derived macrophages reported by Li et al. [44]. For M2-stimulated macrophages, most of the enriched categories were related to the down-regulation of inflammation, such as reduced responses to humoral immunity, neutrophil-mediated immunity, and granulocyte activation (Figure 6B and Appendix A). 

Interestingly, our results strongly indicate the negative effects of NBF on overall cellular activities and energy metabolism. For the NBF condition, the top enriched categories were related to down-regulation of protein translation and mitochondrial metabolism, revealing overall down-regulation of cellular activity and energy metabolism of the cells treated with NBF (Figure 6C and Appendix A). It is possible that some components in the NBF inhibit the metabolic pathways for both oxidative phosphorylation and glycolysis. For example, representative DAMP HMGB1 was reported to inhibit oxidative phosphorylation of colorectal cancer cells [45]. In addition, endogenous DAMPs oxidized phospholipids could suppress both mitochondrial respiration and aerobic glycolysis of mouse macrophages [46]. These results may be important for explaining impaired repair processes seen in older children and adults with femoral head osteonecrosis where healing is very slow and incomplete. In fact, most patients show a lack of healing of the necrotic region of the femoral head and subsequent collapse of the necrotic region leading to osteoarthritis. Since NBF suppressed overall cellular activities and energy metabolism of macrophages, it may explain the slow and incomplete repair process of clearing necrotic debris and healing of the necrotic bone environment. 

Immunometabolism has become a popular topic in recent years since more and more findings were discovered that intracellular metabolic pathways are involved in the regulation of immune cells [47]. The polarization and activation of macrophages are closely linked to the changes in cells’ energy production and metabolism [48]. On the other hand, macrophages’ metabolism also adapts the cells’ differentiation and activation to fulfill their pro- or anti- inflammatory functions [49]. Therefore, elucidating the metabolic change in macrophages will help to define the directions of cells’ differentiation and functions. However, the studies on cell energy metabolism of porcine macrophage or in the context of femoral head osteonecrosis are very limited. Many studies on chronic inflammation using mouse models found that pro-inflammatory macrophages had low OCR but high ECAR, which indicated reduced oxidative phosphorylation but enhanced glycolysis [24,50,51]. Nevertheless, Vijayan et al. reported that the metabolic responses to LPS are different between human and murine macrophages [52]. In addition, similar to our findings in porcine, human monocyte-derived macrophages showed elevated both mitochondrial respiration and glycolysis activity and capacity under GM-CSF induced pro-inflammatory condition compared to M-CSF induced anti-inflammatory condition [53]. 

In addition to transcriptomic profiling, we investigated macrophage metabolic responses to NBF exposure using multiple functional metabolic assays. The Seahorse Cell Mito Stress Test demonstrated that NBF condition led to a marked suppression of both mitochondrial respiration and glycolytic capacity, indicating a global inhibition of energy metabolism (Figure 7). This metabolic down-regulation was further supported by ^13^C-glucose tracing analysis, which showed that although M1 polarization led to increased glycolytic flux and altered TCA cycle metabolite labeling, NBF condition maintained a metabolic profile comparable to Con (Figure 8). Moreover, mitochondrial ROS production staining revealed elevated mitochondrial ROS production in M1 macrophages, but NBF condition did not differ significantly from Con in mitochondrial ROS levels, consistent with a suppressed or quiescent mitochondrial state (Figure 9). These findings align with previous studies suggesting that metabolic rewiring plays a crucial role in macrophage polarization and function [48,51]. The observed reduction in mitochondrial activity and ROS generation in our NBF condition suggests that NBF does not drive a classical inflammatory phenotype, but instead induces a metabolically inert or suppressed state in macrophages, potentially impairing their capacity for immune activation, efferocytosis, or tissue repair in the necrotic bone environment. 

Phagocytosis is a key function of macrophages that helps clear dead cells and promote tissue repair, especially in damaged environments like osteonecrosis. In our study, M2-stimulated macrophages showed the highest phagocytic activity, supporting their known role in anti-inflammatory responses and tissue healing [28,54]. In contrast, M1-stimulated macrophages had reduced phagocytosis, which aligns with their primary role in producing inflammatory signals rather than clearing debris. Interestingly, macrophages treated with NBF also showed low phagocytic activity, similar to M1 cells. This suggests that the necrotic bone environment may impair macrophage phagocytosis, potentially contributing to delayed healing. The reduced function seen in NBF condition may reflect the suppression in energy metabolism and mitochondrial activity, which are important for supporting active processes like phagocytosis.

In this study, we were surprised that NBF did not induce pro-inflammatory gene expression and cell apoptosis process in the porcine PBMC-derived macrophages (Figure 1, Figure 2, Figure 3, Figure 4 and Figure 7A). One possible reason for this is that NBF is a supernatant derived from centrifuging saline wash solutions of the necrotic bone. As a result, it does not contain non-soluble components of the necrotic bone microenvironment, like the necrotic cell debris, necrotic fat, and bone matrix proteins, which may be required for a robust pro-inflammatory response from macrophages. Another possible reason may be that the necrotic bone immune response involves the coordination of several immune cell types with crosstalk among different cell types to escalate the inflammatory response [55]. Thus, NBF stimulation of macrophages alone in the in vitro culture condition may not simulate the necrotic bone environment with the presence of various immune cells. 

In conclusion, this study represents the first of its kind, seeking to determine the transcriptomic and functional responses of porcine macrophages culture using the artificial NBF. In contrast to our initial hypothesis, the NBF-stimulated macrophages did not show the expected pro-inflammatory phenotype. Instead, there was overall down-regulation of cellular activity and energy metabolism. This suggests that the macrophage activity and function in the necrotic femoral head may be impaired in the necrotic bone microenvironment. Additional studies are planned to confirm macrophage phenotype and activity using the porcine model of ischemic osteonecrosis in vivo, since the gene expression profile might be different compared to in vitro expression [56]. These findings highlight a previously unrecognized mechanism by which necrotic bone microenvironments may suppress innate immune function and impede the resolution of inflammation in ischemic osteonecrosis. Further studies are needed to confirm macrophage activity in response to the non-soluble necrotic bone particle and explore the responses of other types of immune cells in the necrotic bone environment.

## Figures and Tables

**Figure 1 cells-14-01258-f001:**
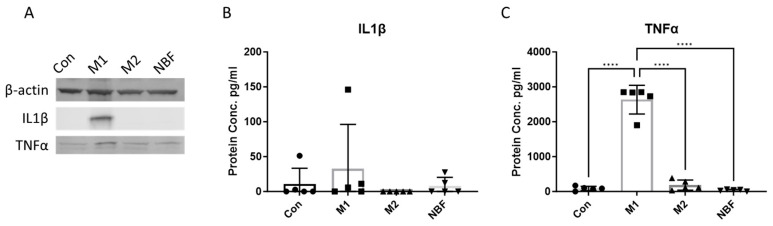
The protein expression of pro-inflammatory cytokines. (**A**). The western results of the expression of IL1β and TNFα among Con, M1, M2, and NBF conditions in cell lysates (**B**). The ELISA results of the comparison of IL1β level among Con, M1, M2, and NBF conditions in cell culture supernatants. (**C**). The ELISA results of comparison of TNFα level among Con, M1, M2, and NBF conditions in cell culture supernatants. (****: *p* < 0.0001).

**Figure 2 cells-14-01258-f002:**
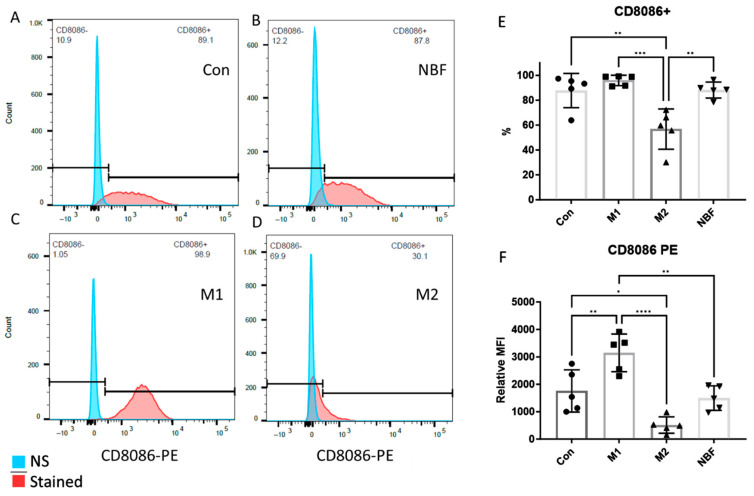
The FACS results of the cell surface marker CD8086. (**A**–**D**). Histograms of the CD8086 signal density of Con, NBF, M1, and M2 vs. non-stained (NS) sample. (**E**). The comparison of the percentages of the CD8086 positive cells among Con, M1, M2, and NBF conditions. (**F**). The relative median fluorescence intensity (MFI) comparison among Con, M1, M2, and NBF conditions. (*: *p* < 0.05, **: *p* < 0.01, ***: *p* < 0.001, ****: *p* < 0.0001)).

**Figure 3 cells-14-01258-f003:**
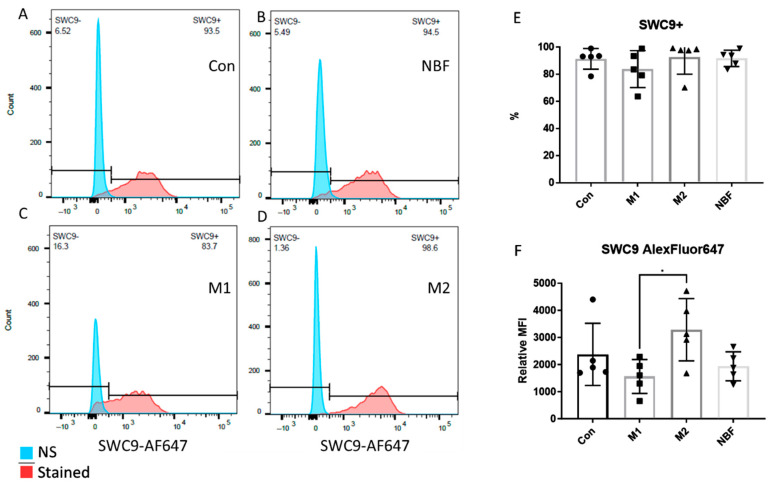
The FACS results of the cell surface marker SWC9. (**A**–**D**). Histograms of the SWC9 signal density of Con, NBF, M1, and M2 vs. non-stained (NS) sample. (**E**). The comparison of the percentages of the SWC9 positive cells among Con, M1, M2, and NBF conditions. (**F**). The relative median fluorescence intensity (MFI) comparison among Con, M1, M2, and NBF conditions (*: *p* < 0.05).

**Figure 4 cells-14-01258-f004:**
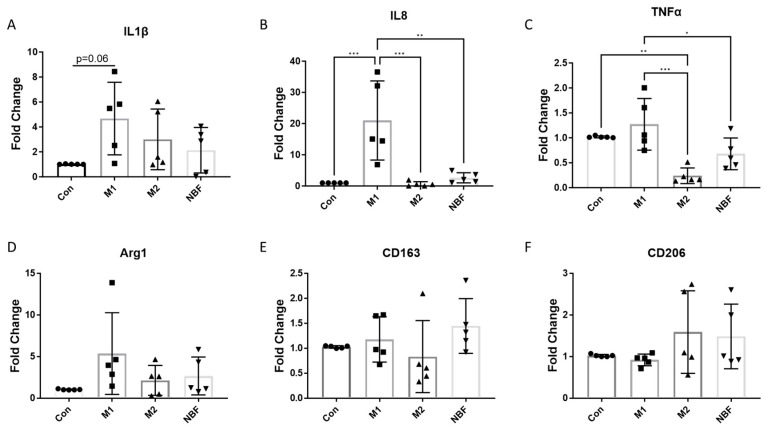
The RT-qPCR results of the pro- and anti-inflammatory genes. (**A**–**F**). The comparisons of the RNA expressions of IL1β, IL8, TNFα, Arg1, CD163, and CD206 among Con, M1, M2, and NBF conditions (*: *p* < 0.05, **: *p* < 0.01, ***: *p* < 0.001).

**Figure 5 cells-14-01258-f005:**
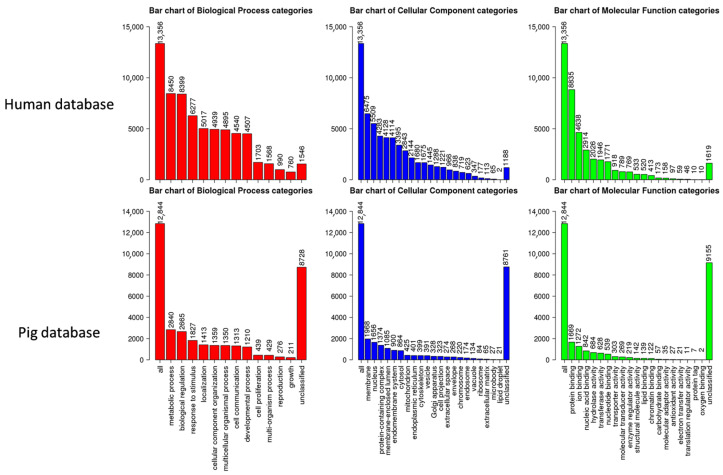
GO Slim summary figures used human database (**top panel**) and pig database (**bottom panel**).

**Figure 6 cells-14-01258-f006:**
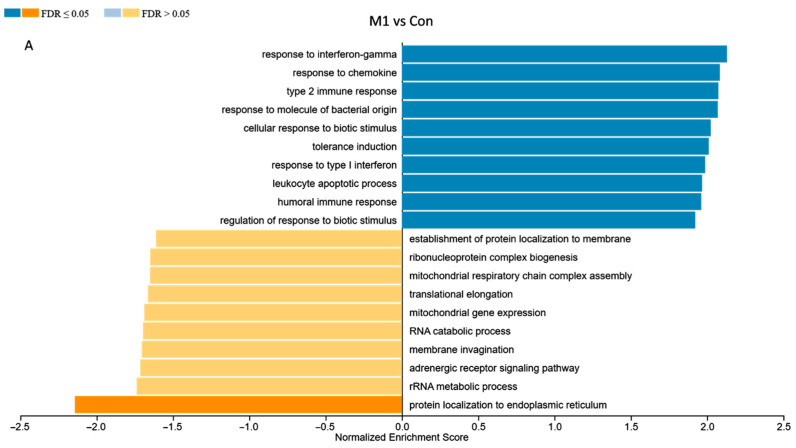
Gene Set Enrichment Analysis (GSEA) of the bulk RNA sequencing results. (**A**). M1 vs. Control (Con) conditions. (**B**). M2 vs. Con conditions. (**C**). NBF vs. Con treated conditions.

**Figure 7 cells-14-01258-f007:**
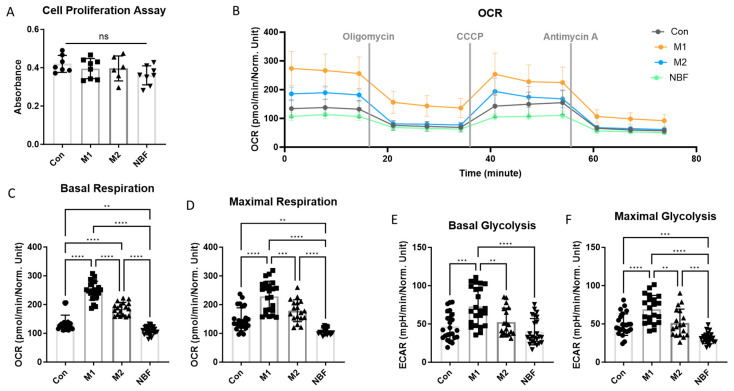
Seahorse Cell Mito Stress Test. (**A**). The comparison of the results of the Cell Proliferation Assay among Con, M1, M2, and NBF conditions. (**B**). The dynamic change in the oxygen consumption rate (OCR) of cells among Con, M1, M2, and NBF conditions. (**C**). The comparison of basal respiration rate among Con, M1, M2, and NBF conditions. (**D**). The comparison of maximal respiration rate among Con, M1, M2, and NBF conditions. (**E**). The comparison of basal glycolysis rate among Con, M1, M2, and NBF conditions. (**F**). The comparison of maximal glycolysis rate among Con, M1, M2, and NBF conditions (**: *p* < 0.01, ***: *p* < 0.001, ****: *p* < 0.0001).

**Figure 8 cells-14-01258-f008:**
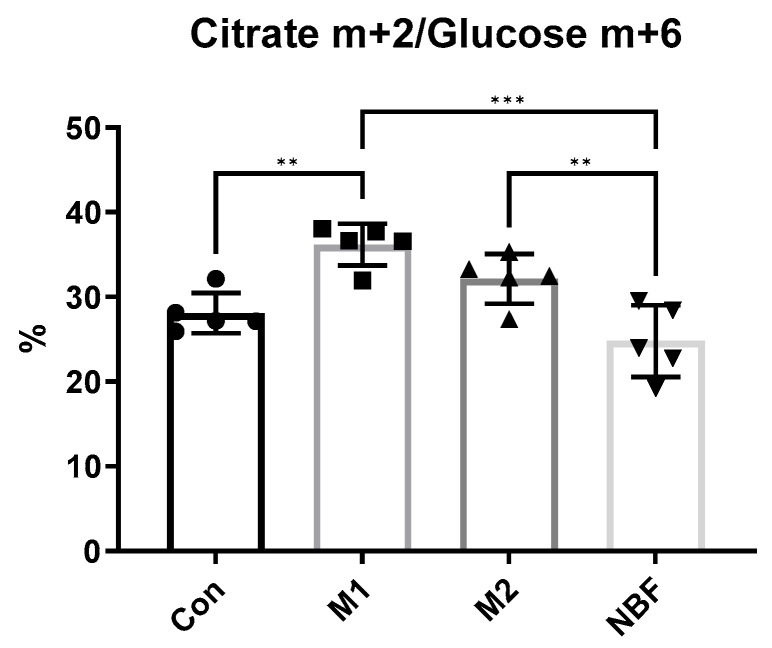
Metabolic Flux Analysis. The comparison of the percentages of citrate m+2 out of glucose m+6 in cells cultured in [U-^13^C_6_] glucose medium among Con, M1, M2, and NBF conditions (**: *p* < 0.01, ***: *p* < 0.001).

**Figure 9 cells-14-01258-f009:**
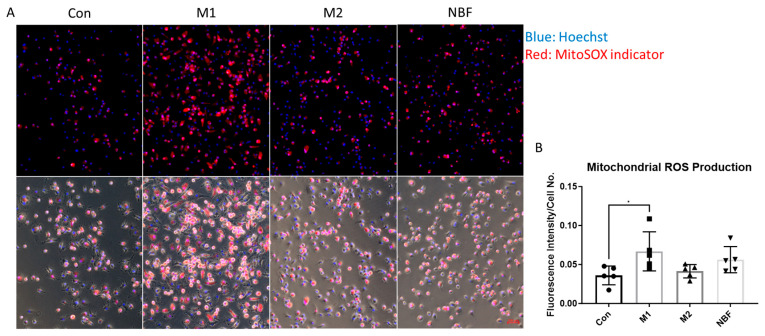
MitoSOX^TM^ live cell staining. (**A**). The representative images of merged fluorescence channels (top panel; blue: Hoechst: red: MitoSOX indicator) and merged with bright field (bottom) of macrophages cultured in Con, M1, M2, and NBF conditions. (**B**). The comparison of the MitoSOX^TM^ fluorescence intensity normalized to cell number among Con, M1, M2, and NBF conditions (*: *p* < 0.05).

**Figure 10 cells-14-01258-f010:**
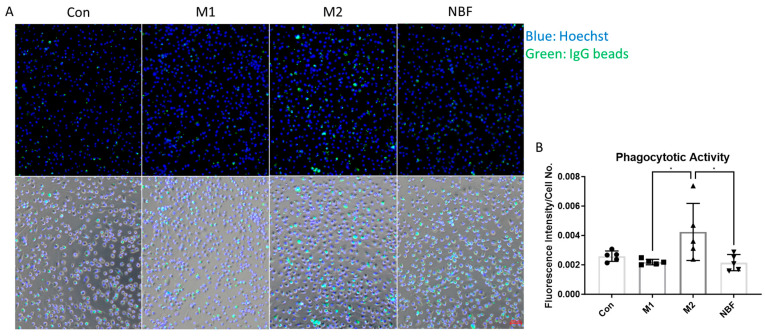
Phagocytosis Assay. (**A**). The representative live cell images of merged fluorescence channels (top panel; blue: Hoechst: green: FITC IgG beads) and merged with bright field (bottom) of macrophages cultured in Con, M1, M2, and NBF conditions. (**B**). The comparison of the FITC IgG fluorescence intensity normalized to cell number among Con, M1, M2, and NBF conditions (*: *p* < 0.05).

**Table 1 cells-14-01258-t001:** Quantitative RT-PCR primer list.

Gene Name	Sequence	
IL1β	Forward:	ACCAGGGTTACTGACTATGGC
	Reverse:	GTTGAGGCAGGAAGGAGAT
IL8	Forward:	AGCCACGGAGAATGGGTTTT
	Reverse:	TGGGTGCAGAAGAAGGTTGT
TNFα	Forward:	CGCATCGCCGTCTCCTACCA
	Reverse:	GCCCAGATTCAGCAAAGTCCAGAT
Arg1	Forward:	AGCCCAGCAAGTTCATACCT
	Reverse:	ACCAGCCAGCTTTGTCAGAT
CD163	Forward:	TGCCATGAAGAGGGTAGGGT
	Reverse:	GTCTTGCCATTCACCAAGCG
CD206 (Mrc1)	Forward:	AGCATCAGGAAAGGACCAGC
	Reverse:	GGCAACCGGAAGGAAAGAGA
GAPDH	Forward:	ACACTCACTCTTCTACCTTTG
	Reverse:	CAAATTCATTGTCGTACCAG

## Data Availability

The bulk RNA sequencing data are available via Sequence Read Archive (SRA) database (accession number: PRJNA1213853).

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
