# Peer review of "Necrotic Bone Fluid Suppresses Energy Metabolism of Porcine PBMC-Derived Macrophages In Vitro"

_cells, 2025, doi:10.3390/cells14161258_

Round 1
Reviewer 1 Report
Comments and Suggestions for Authors
Title:
The title is fine but it should be more informative. It doesn't show the potential applications or the novelty of the findings. For example, the significant effects and study scope (in vitro, ex vivo, …) should be clear.
Introduction:
(1) Although the authors in the introduction mentioned M1 and M2 macrophages, it should be more detailed, explaining their roles in bone healing and how their dysfunction might contribute to osteonecrosis. (2) Moreover, the authors should mention why macrophage energy metabolism is crucial in this process. Explain more how metabolic shifts affect macrophage function (cytokine production, phagocytosis) and polarization and cite properly in the text. (3) Add more recent reviews on macrophage immunometabolism and osteonecrosis.
Methods:
The methods section is well-detailed, and well organized. However, some sections should be improved or added; (1) Regarding the assessment of the differentiation and maturation, you should include information to show how macrophage differentiation was confirmed (by morphological changes, expression of specific macrophage markers). (2) Regarding the flow cytometry for macrophage markers, why did you only analyze CD80/86 and SWC9 expression, and did not include other macrophage markers (CD68, CD163)? If possible, use CD163 which is more important in confirming M2 polarization. (3) MAJOR: To confirm the current findings, additional experiments are needed and I suggest the researchers to accomplish in vitro phagocytosis assay to assess the effect of NBF on macrophage phagocytic activity, and western blot, to confirm the protein expression results from ELISA (at least for selected proteins). (4) MAJOR: Although the Seahorse assay gives us valuable information on cellular energy metabolism, additional experiments should be done to confirm metabolic changes by NBF. Metabolic flux analysis (Tracing labeled metabolites such as glucose, and glutamine through different metabolic pathways), reactive oxygen species (ROS) production assay, measuring the expression and activity of key enzymes involved in glycolysis (hexokinase, pyruvate kinase) and mitochondrial respiration (citrate synthase, cytochrome c oxidase), and also evaluating Lipid Metabolism are suggested. (5) Regarding statistical analyses correlation analyses between quantitative variables (gene expression, protein levels, metabolic activity) could provide some overview and added information about the relationships between these variables.
Results:
(1) When describing the FACS results, the gating strategy should be included in the pictures and explained in the figure legend. The histograms are informative but showing the gating is necessary. Moreover, summarize the key findings in the figure legend.
Discussion and Conclusion:
(1) You should consider adding more discussion of the limitations of the study, by including limitations related to the in vitro model, the use of artificial NBF, and the focus on a single cell type. (2) Since you have mentioned that your findings are different from other studies that have shown increased glycolysis in pro-inflammatory macrophages, you should provide more potential reasons and mechanistic studies for these discrepancies.
Author Response
Title:
The title is fine but it should be more informative. It doesn't show the potential applications or the novelty of the findings. For example, the significant effects and study scope (in vitro, ex vivo, …) should be clear.
Response: Title was corrected according to the suggestion.
Introduction:
(1) Although the authors in the introduction mentioned M1 and M2 macrophages, it should be more detailed, explaining their roles in bone healing and how their dysfunction might contribute to osteonecrosis. (2) Moreover, the authors should mention why macrophage energy metabolism is crucial in this process. Explain more how metabolic shifts affect macrophage function (cytokine production, phagocytosis) and polarization and cite properly in the text. (3) Add more recent reviews on macrophage immunometabolism and osteonecrosis.
Response: More information was added in Introduction.
Methods:
The methods section is well-detailed, and well organized. However, some sections should be improved or added; (1) Regarding the assessment of the differentiation and maturation, you should include information to show how macrophage differentiation was confirmed (by morphological changes, expression of specific macrophage markers).
Response: We added “cells were differentiated from monocyte to macrophages as most cells were firmly attached to the cell culture plate”, highlighted in the metho. We also observed over 80% of cells expressed macrophage maturation marker SWC9.
(2) Regarding the flow cytometry for macrophage markers, why did you only analyze CD80/86 and SWC9 expression, and did not include other macrophage markers (CD68, CD163)? If possible, use CD163 which is more important in confirming M2 polarization.
Response: We tried, but we could not find a sensitive porcine CD163 antibody for flow cytometry.
(3) MAJOR: To confirm the current findings, additional experiments are needed and I suggest the researchers to accomplish in vitro phagocytosis assay to assess the effect of NBF on macrophage phagocytic activity, and western blot, to confirm the protein expression results from ELISA (at least for selected proteins).
Response: Western and phagocytosis assay results were added.
(4) MAJOR: Although the Seahorse assay gives us valuable information on cellular energy metabolism, additional experiments should be done to confirm metabolic changes by NBF. Metabolic flux analysis (Tracing labeled metabolites such as glucose, and glutamine through different metabolic pathways), reactive oxygen species (ROS) production assay, measuring the expression and activity of key enzymes involved in glycolysis (hexokinase, pyruvate kinase) and mitochondrial respiration (citrate synthase, cytochrome c oxidase), and also evaluating Lipid Metabolism are suggested.
Response: MitoSOX and 13C metabolic flux analysis were added.
(5) Regarding statistical analyses correlation analyses between quantitative variables (gene expression, protein levels, metabolic activity) could provide some overview and added information about the relationships between these variables.
Response: We added more assays in the paper according to your suggestion. However, we believe it’s not appropriate to do correlation analyses among all these assays since most of them are not directly related.
Results:
(1) When describing the FACS results, the gating strategy should be included in the pictures and explained in the figure legend. The histograms are informative but showing the gating is necessary. Moreover, summarize the key findings in the figure legend.
Response: Gating strategy was added to the Methods section.
Discussion and Conclusion:
(1) You should consider adding more discussion of the limitations of the study, by including limitations related to the in vitro model, the use of artificial NBF, and the focus on a single cell type. (2) Since you have mentioned that your findings are different from other studies that have shown increased glycolysis in pro-inflammatory macrophages, you should provide more potential reasons and mechanistic studies for these discrepancies.
Response: More discussion paragraphs were added. The limitation and possible reasons were listed in the discussion section.
Reviewer 2 Report
Comments and Suggestions for Authors
1 All the experiment designed base on the cells, why choose the big animal model (pig).
2 artificial NBF is the similar with the NBF in vivo? It will be persuasive that the paper provides the NBF works in animal model not only by cell models in intro.
Author Response
1 All the experiment designed base on the cells, why choose the big animal model (pig).
Response: Our lab has established pig Perthes disease model which is closer to human patient compared to mouse osteonecrosis model. Even though all experiments in this study were in vitro, we believe it better represents of human PBMC derived macrophages.
2 artificial NBF is the similar with the NBF in vivo? It will be persuasive that the paper provides the NBF works in animal model not only by cell models in intro.
Response: In our previous paper published in Bone (Damage associated molecular patterns in necrotic femoral head inhibit osteogenesis and promote fibrogenesis of mesenchymal stem cells; PMID: 34571205), we compared artificial NBF with true NBF (extracted from in vivo necrotic bone), the DAMP components and effects on MSCs were similar.
Round 2
Reviewer 1 Report
Comments and Suggestions for Authors
Thank you for your revised manuscript and for the detailed responses to the comments. I appreciate the significant effort you have put into addressing the points raised. You have successfully implemented a number of the key suggestions. There are a few points from the initial review that were either not addressed. However, the manuscript is now suitable for publication and I am convinced now with your replies.
Sincerely,